# Specialty Coffee Shops in Mexico: Factors Influencing the Likelihood of Purchasing High-Quality Coffee

Roselia Servín-Juárez [1],*, Carlos J. O. Trejo-Pech [2], Alma Yanet Pérez-Vásquez [1] and Álvaro Reyes-Duarte [3]

1 Campus Córdoba, Colegio de Postgraduados, Amatlán de los Reyes, Veracruz 94946, Mexico; perez.alma@colpox.mx
2 Department of Agricultural & Resource Economics, University of Tennessee at Knoxville, Knoxville, TN 37996, USA; ctrejope@utk.edu
3 Escuela de Agronomía, Facultad de Recursos Naturales y Medicina Veterinaria, Universidad Santo Tomás, Santiago 8370003, Chile; areyes@santotomas.cl
* Correspondence: roseliasj@colpos.mx

**Abstract:** This study models the purchasing behavior of specialty coffee by 114 coffee shops across 15 cities in nine states in Mexico. Simple and multilevel mixed-effects logistic models are tested. Our models extend the framework used in prior research. We model the purchase of specialty coffee as a function of: (a) material attributes, (b) symbolic attributes, (c) coffee shop characteristics, (d) profile of the coffee shop's owner, and (e) socio-economic variables of the cities where the coffee shops were located. Overall, our results are consistent with expectations developed from the coffee literature. That is, the likelihood of purchasing specialty coffee increases when: coffee's aroma drives the purchase, coffee purchased is from the state of Oaxaca, the coffee shop has a value-added business model, the coffee shop is diversified selling both ground coffee and coffee drinks, the coffee shop owner's knowledge on coffee supply chain activities is high, and the coffee shop is located in a city with a higher education index. In contrast, the likelihood of purchasing specialty coffee decreases when a coffee professional tastes the coffee before the purchase, when coffee shops are not given the opportunity to roast their own coffee, and in coffee shops located in larger cities. Overall, our research suggests that the specialty coffee niche in Mexico has some elements required for this segment to transition from a supply chain approach to a value-based supply chain approach. This might be particularly beneficial for smallholder coffee growers, who despite several constraints contribute to the sustainability of coffee supply chains.

**Keywords:** coffee shops; coffee quality attributes; specialty coffee; coffee from Mexico; sustainability of small-scale coffee farmers; value-based supply chain

## 1. Background

### 1.1. Introduction

Following the international market liberalization of coffee in the late 1980s, global coffee production and price volatility increased and overall product quality declined [1]. As a reaction to the decline in coffee quality, the concept of specialty coffee was introduced in the coffee industry. Starbucks® initiated the commercialization of specialty coffee in the United States in the 1980s, marking a turning point in coffee consumption. Starbucks® offered a more artisanal beverage prepared with better beans, immediate extraction, advanced preparation techniques, and more personalized interaction with the client [2]. All of this new dynamic inspired a generation of young coffee entrepreneurs [3,4].

Specialty coffee emerged as a niche market where consumers perceive sensory attributes such as taste, aroma, body, sweetness, and bitterness as distinctive product attributes [5]. The Specialty Coffee Association of America (SCAA) defines high-quality coffee or Q Certified Coffee® as one with a quality score of more than 80 cup points out of

100 when evaluated following the SCAA's quality protocols [6]. More specifically, high-quality coffee with 80 to 84 points is graded as premium coffee and high-quality coffee with more than 85 as specialty coffee. This study analyzes coffee shops in Mexico purchasing high-quality coffee and focuses particularly on those coffee shops buying specialty coffee.

In Mexico, specialty coffee shops emerged in the 1990s when young entrepreneurs tried to mimic the success of Starbucks®. The pioneers in Mexico were Coffee Factory®, followed by franchises such as Café Etrusca®, Coffee House®, and Gloria Jean's Coffees®. Mexican consumers embraced relatively quick the emergence of local coffee shops. Euromonitor International [7] indicates that the economic value of café/bars represented around 9% of the total value of the consumer foodservice sector during the 2005–2019 period in Mexico. In 2019 alone, cafés/bars' transactions value and number of outlets grew 6%, with the number of outlets reaching 32,868. The specialty coffee shop market, a segment of the cafés/bars food service industry, grew annually between 20% and 25% during the same period.

The growth of specialty coffee shops in Mexico might significantly increase the demand for coffee and improve the economic conditions in the producing regions [8]. Specialty coffee shops create an alternative market for smallholder coffee farmers, who otherwise sell their crops to intermediaries or to the export market [2], outlets that do not necessarily pay a price premium for high-quality coffee. A still limited specialty coffee market outlet is the Cup of Excellence, a coffee auction where high-quality coffee producers can offer their coffee at prices up to around four to five times greater than in conventional markets [9]. Further, the participation of smallholder coffee growers in the specialty coffee market contributes to the sustainability of coffee supply chains. The involvement of smallholder coffee growers in this market segment provides desired socio-economic, technological, and environmental benefits for coffee producers and reduces the communities' cultural vulnerabilities [10–12].

The goal of this study is to identify the factors influencing the purchase of specialty coffee by coffee shops in Mexico. This can benefit more sustainable smallholder coffee farmers selling coffee to coffee shops directly or indirectly through coffee cooperatives by providing them with relevant information to better target their customers. Sustainable small shareholder coffee supply chains can be economically sustained since consumers are willing to pay a higher price for specialty coffee compared to conventional coffee grown on larger industrial plantations [13,14]. These types of coffee supply chains can also be more sustainable from both environmental and community perspectives. For example, smallholder coffee farmers in Colombia work directly with buyers, roasters, and/or importers to achieve desired environmental, socio-economic, technological and community benefits. Smallholder coffee supply chains working closely with other players of the supply chain are associated with sustainable production characteristics such as higher tree diversity for shade-grown systems, improved water and soil conservation, and organic management. Smallholder coffee production systems can also empower farmers to better understand and operate within these types of supply chains [11]. Research has indirectly linked high-quality coffee systems and sustainability as well. Recent meta-analyses show smallholder coffee farmers producing certified, high-quality coffee receive more positive than negative impacts across environmental, human, social, natural, and economic dimensions of sustainability [15,16]. Mexico is a relevant coffee producing region supplying certified/high-quality coffee markets [15–17] where there are about 500,000 small coffee farms averaging 1.4 ha/farm which employ more than 3 million workers mostly from poor indigenous communities [15].

This study elicits from coffee shop owners the factors that are relevant when buying specialty coffee as an input for their coffee beverages. Material and symbolic quality attributes of specialty coffee play a role in the buying process [12,18]. For instance, the price of specialty coffee bought on an eBay style auction open to the general public was influenced by material and symbolic attributes [18]. However, the process of purchasing specialty coffee as an input for coffee beverages by coffee shops remains unexplored. To

fulfill this gap in the literature, we model the probability of coffee shops buying specialty coffee based on material and symbolic quality attributes. Material attributes include features resulting from biological, physical, or chemical processes and can be measured using human senses. Symbolic attributes are based on concepts that differentiate the product, reduce information asymmetry, and create value; and include business reputation and prestige such as trademarks, geographic origin, and sustainability practices.

In addition to material and symbolic quality attribute variables, we include in the model characteristics of the coffee shops and of the socio-economic context in which the shops are located. Coffee shop's characteristics include age of coffee shop, age of its customers, whether the coffee shop operates in an attractive location, among others. Socio-economic variables include the number of inhabitants and level of education in the cities where the coffee shops are located.

To recap, the model proposed in this study assumes that a specialty coffee shop buyer/owner decides the level of quality coffee to buy based on his/her personal background, material and symbolic attributes, specific coffee shop characteristics, and socio-economic conditions. This is the first study to model the demand-side of specialty coffee shops in Mexico. The analysis uses primary data collected from 114 coffee shop owners/buyers across nine states of Mexico and 18 cities (Figure 1).

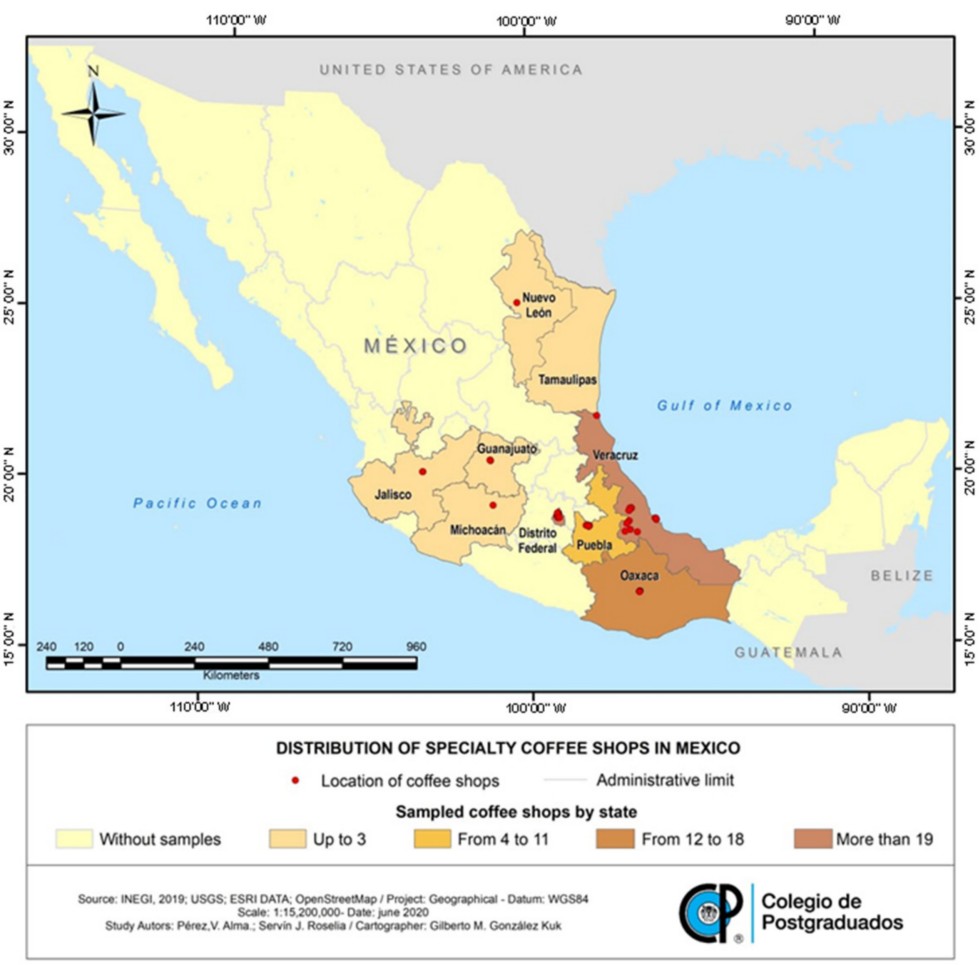

**Figure 1.** Distribution of specialty coffee shops in Mexico.

## 1.2. Specialty Coffee Quality

The concept of quality in mass consumer products has different connotations. To delimit the various meanings of quality, studies have followed two approaches [19,20]. The first approach defines quality according to a set of product specifications [21]. The quality attributes of raw material become relevant on defining the quality of the final product [5,18].

A second approach has to do with the user's expectations and considers the difference between the product's attributes and the customer's needs, i.e., how well the products meet expectations [22]. Under this expectations-based approach, other attributes such as symbolic attributes that customers recognize in the product they wish to buy, are relevant.

Another perspective of quality in mass consumption products has been suggested [23]. From this perspective, high quality is associated with high specification standards. A high-quality product or service is evaluated in absolute terms. The value of the material or service is of high quality, measured as perfection in detail, reliable performance, or generating a clear emotional and status appeal. For products with high specification standards, cultural issues influence the perceived quality of a product or service in aspects such as self-directed symbolic values, symbolic values directed by others, experience values, utilitarian values, and financial values [24]. Therefore, high specification products are intended to meet exigent demanding customers' needs. Under this perspective, aspects such as the in-person service, store characteristics, and socio-economic variables are relevant [5,18].

Quality sensory analysis is associated with coffee quality [19]. Sensory analysis includes systematized procedures that are based on human senses, with strict classifications developed by experts, which guide evaluations and classify the products in a quality ranking. In most cases, this analysis uses scorecards, rankings, and paired tests to bring more accuracy and reliability to the evaluation. This quality evaluation procedure is used, in addition to coffee, in products such as wine, whisky, tequila, olive oil and tea.

The SCAA developed a precise meaning for specialty coffee quality [25]. The term "specialty coffee" refers to a coffee that can reach a grade equal or above 85 cup points under the SCAA scale. Evaluated sensory attributes of coffee include fragrance/aroma, flavor, aftertaste, acidity, body, uniformity, balance, clean cup, sweetness, and overall attribute [26,27]. Either a Q grader®, a coffee specialist certified by the Coffee Quality Institute® [28] or a coffee professional must perform the sensory analysis (we use the term coffee professional hereafter to refer to Q graders or other types of coffee professionals evaluating the quality of coffee purchased). A Q grader is a person who received training in cupping of coffee to identify specialty quality considering the internationally accepted physical and sensory scoring system of the SCAA.

The specialty coffee industry has adopted another term to further distinguish specialty coffees: "boutique" coffee. Boutique coffees are the modern equivalent of the specialty coffees of the late 1980s and early 1990s as they are distinguished and valued for their refined flavor, unique growing region, and especially their limited availability [29,30]. Procurement of boutique coffees is often challenging and farmers who grow these coffees must seek out buyers willing to pay adequate premiums for quality.

## 2. Hypotheses: Factors Influencing the Purchase of Specialty Coffee

### 2.1. Material Attributes

Sensory analysis in coffee is based on the product's physical and chemical properties. The coffee bean's appearance, flavor, and aroma define its quality [31]. In turn, the quality of a coffee drink is mainly affected by the chemical composition of the coffee bean [32,33]. This composition is determined by multiple factors including the crop genetics, altitude of location where the coffee is produced (altitude hereafter), agricultural practices, harvesting, and post-harvesting practices. In this research, we consider a variable indicating whether coffee professionals performed a sensory test before purchasing coffee (professional) and the variables roast, aroma, and altitude as potential material attributes influencing the purchase of specialty coffee.

Coffee quality evaluation following the SCAA protocol is costly. Hence, coffee shops might decide to buy coffee without following formal evaluation protocols but based only on the informal judgment of a coffee professional. Coffee professionals may reduce costs by purchasing high-quality coffee graded as premium coffee (i.e., with 80 to 84 cup points) rather than specialty coffee (i.e., above 85 points), and still bring high-quality coffee to prepare coffee blends that meet customer preferences. On the other hand, performing

formal coffee quality evaluations might allow coffee shops to purchase better quality coffee beans and, consequently, offer their consumers premium blends at higher prices. Therefore, the expected sign of variable professional remains an empirical question.

The quality and type of roasted coffee and coffee's aroma are attributes valued by specialty coffee [34–36]. We expect aroma to positively affect the buying decision of specialty coffee by coffee shops [36–41]. Coffee quality is highly sensitive to roast processing, storage, and transportation [12], thus coffee shop owners/buyers were asked whether roasted coffee availability drives their purchase decision. Given that about one half of coffee shops in our sample roast their own coffee, the direction of the prediction for roast is unknown. Altitude has been used as a proxy for high-quality coffee. The higher the altitude, the better the quality attributes evaluated in a coffee beverage [42]. For instance, coffees from plantations between 1550 and 1780 m above the sea level in Costa Rica have better body, typicity and are in general preferred by consumers over coffees produced in lower altitudes [43]. In addition, altitudes above 1219 m increase the percentage of dry weight of the chemical coffee components including caffeine, trigonelline, fat, sucrose, and chlorogenic acids. In addition, roasters and importers often market coffees to be "mountain" or "high grown" coffees to signal high quality [5,30]. Therefore, we expect altitude to be an attribute positively influencing specialty coffee purchasing behavior.

### 2.2. Symbolic Attributes

We consider explanatory variables indicating whether the coffee shops differentiate their businesses based on two concepts: the history of the coffee shop (i.e., its legacy in society) and/or a value-added business model. Adopting value-added based thinking implies the understanding that it is not only the material quality attributes that coffee shops and consumers are buying but also the symbolic attributes and personalized service [44]. Specialized coffee shops compete by establishing a personal connection with the consumer and providing an environment of exclusivity, targeting clients who have refined tastes and know-how to appreciate flavor [14]. In a typical coffee shop, the cost of coffee as raw material represents only a very small portion of total cost. What all this suggests is that coffee shops are value-adding mechanisms, and coffee is only the "great pretext" that explains the existence of coffee shops [45]. It is plausible that coffee shop owners with a clear business concept focus on history or value-added value coffee quality differently than other coffee shops. For instance, focused coffee shop owners might be more interested in matching the concept they developed to differentiate their businesses than in acquiring coffee of the highest quality. However, the direction of the relationship is difficult to predict because coffee shops with a clearly defined business models might also value specialty coffee. The latter scenario might be the case of top performing coffee shops.

We also consider location variables related to the origin of coffee. Coffee beans produced in the states of Oaxaca and Guerrero are associated with high-quality coffee and traditional taste that is well perceived by Mexican consumers. Thus, we expect a positive relationship between coffees grown in these states and the probability of buying specialty coffee by coffee shops.

In Oaxaca, a competition and auction event, the Oaxaca Coffee Quality Award, has been implemented from 2016 to 2020 to commercialize specialty coffee lots of small and medium farms. In these events, national buyers acquired 64% of the micro-lots at an average price of US$ 12.85 per kg ($5.83/lb). Three international buyers acquired 36% of the remaining micro-lots at an average price of US$ 13.76 per kg ($6.24/lb). The vast majority of the winning producers established direct supply relationships with buyers of specialty coffee, who offered them higher prices than the ones usually received when selling their coffee through conventional marketing channels, although lower than the ones of the auction [45]. In addition, the coffee produced in Guerrero is famous at the national level for its high quality. Nine coffee farmers participated in the Cup of Excellence coffee auction, organized by Asociación de la Cadena Productiva del Café. A total of 6.39 metric

tons of coffee were commercialized through this channel, with an average quality score of 87.7 cup points [46].

### 2.3. Coffee Shop Characteristics

The years of experience in the coffee business might help coffee shops better identify the coffee quality they buy without the need to perform a sensory analysis in a laboratory or by a certified cupper [47]. A positive relationship between our control variable age_shop and specialty coffee acquisition is expected.

In the restaurant sector, geographic location is relevant for market segmentation, and customers value differently various attributes depending on store locations [48,49]. Many consumers visit coffee shops to rest when they are shopping [50]. Thus, if coffee shops are visible, clients are more likely to visit and buy their coffee, keeping product's quality constant. As a consequence, owners of well-located coffee shops (i.e., visible coffee shops) might care less about buying high graded coffee than those located in places not considered highly visible. A negative relationship is expected between coffee shop visibility (visibility) and purchase of specialty coffee.

Coffee shops owners may realize that clients are interested on having the opportunity to visit the establishments at convenient hours. Coffee shop owners aware of this convenience attribute might be willing to have extended opening hours. We expect the more hours the coffee shop works (w_hours), the more likely is to buy specialty coffee. This expectation assumes that the more conscious a coffee shop owner is towards his/her clients, the more conscious would be towards the overall business, and the more the owner would tend to buy specialty coffee [35].

By selling ground coffee rather than in-store coffee drinks only, coffee shops might attract more high-quality oriented customers. High-quality coffee consumers may demand both drink and ground coffee at the coffee shop. Therefore, a positive relationship between variable ground and purchase of specialty coffee is expected. In other words, a higher demand of specialty coffee might be the result of diversification of the supply by coffee shops.

If the coffee shop is also an intermediary company (intermediary) selling coffee to roasters/retailers, the coffee shop owner would likely need to acquire specialty coffee. These type of buyers generally demand high-quality premium products [45,51]. Consumers' age also determines the preferences of drinking coffee [35]. Younger consumers (young_cust) drink relatively more coffee for stimulation than older consumers do. Further, younger consumers prefer specialty coffee to other coffee types [36]. Therefore, we expect a positive relationship between young_cust and specialty coffee purchase.

### 2.4. Coffee Shop Owner Characteristics

Younger coffee shop owners may be more auto determinant, trusting their capacity to provide high-quality drinks to their customers for a given quality level of coffee bean. In contrast, older owners may be more traditional and depend more on the quality score of the coffee bean purchased. Owners' age (age_own) and specialty coffee might be positively related. We also expect a gender (gender_own) effect on specialty coffee. Being a male coffee shop owner might increase the likelihood of purchasing specialty coffee. Most high-quality coffee producers in Mexico are male, and negotiations are more likely to occur smoothly between pairs, male farmers and male shop owners/buyers.

The higher the coffee shop owner's level of education (education_own), the easier is for the owner to gather and analyze market data, and navigate the specialized specialty coffee segment [52,53]. Similarly, the coffee shop owner's level of knowledge (knowledge) on barismo techniques (e.g., cup preparation) and other coffee supply chain practices (e.g., roasting, traceability, and processing) should make owners/buyers more knowledgeable in the acquisition of specialty coffee and in consequence increase their likelihood to purchase this type of coffee.

In coffee shops with a unique owner rather than with various owners, the decision making process is more likely concentrated in the owner. Unique owners might know more people in the supply chain (roasters, baristas, producers, entrepreneurs) compared to owners in coffee shops with multiple owners [54]. This knowledge might increase the possibility of acquiring specialty coffee.

### 2.5. Socio-Economic Characteristics

We also control for socio-economic characteristics of the context on which coffee shops are located [35]. In this respect, we include as control variables the average years of education and the total population of the city where the coffee shop is located. Level of education and consumption of high-quality coffee are directly related [35,55–58]. The direction of the relationship between population size and specialty coffee is difficult to predict.

Table 1 summarizes the predictions and describes the variables. While not directly comparable to our study, variables in this study have been widely used in coffee and other specialty products studies. Table A1 in the Appendix A shows selected studies referring to those variables.

**Table 1.** Variables for the econometric analysis and predictions.

| Variable Categories | Variable | Description | Expected Correlation with Y |
|---|---|---|---|
| Independent | Y | 1 if specialty coffee purchased, 0 else | NA |
| Characteristics influencing the purchasing decision (material) | *Professional* | 1 if coffee tasted by coffee professional, 0 else | +/− |
| | *Roast* | 1 if coffee roast, 0 else | +/− |
| | *Aroma* | 1 if coffee aroma, 0 else | + |
| | *Altitude* | 1 if altitude of coffee production, 0 else | + |
| Coffee shop business models and location of coffee purchased (symbolic) | *History* | 1 coffee history, 0 else | +/− |
| | *Value-Added* | 1 if Value-Added, 0 else | +/− |
| | *Oaxaca* | 1 if purchased from Oaxaca, 0 else | + |
| | *Guerrero* | 1 if purchased from Guerrero, 0 else | + |
| Coffee shop characteristics | *Age_Shop* | Number of years coffee shop operating | + |
| | *Visibility* | 1 if visible/attractive locations, 0 else | − |
| | *W_hours* | Hours open peer week | + |
| | *Ground* | 1 if sells ground coffee, 0 else | + |
| | *Intermediary* | 1 if shop is an intermediary, 0 else | + |
| | *Young_Cust* | 1 if consumers mostly 21–30 years old, 0 else | + |
| Coffee shop owner characteristics | *Age_Own* | Age in years | + |
| | *Gender_Own* | 1 if male, 0 female | + |
| | *Education_Own* | 1 if bachelor's or higher degree, 0 else | + |
| | *Knowledge* | 1 if knowledgeable about roasting, 0 else | + |
| | *Unique_Owner* | 1 if only one owner, 0 if more | + |
| Socio-economic characteristics of city where coffee shop located | *Population* | Total city population | +/− |
| | *Education* | Human development index using literacy & school attendance | + |

Notes: Italics indicate name of variables as used in the regressions. Specialty coffee is defined according to the standards by the Specialty Coffee Association of America (SCAA), which considers specialty coffee those receiving at least 85 points out of 100 when evaluated following the SCAA protocols. Coffees receiving between 80 and 84 points are also considered high-quality coffee with a grade of "premium" but no specialty coffee [6]. Indeed, in this study all coffee shops in the sample buy high-quality coffee (with at least 80 points). The last two indicators were reported by the National System of Statistical and Geographical Information of Mexico (INEGI by its acronym in Spanish) [59].

## 3. Methods

### 3.1. Data

Data were collected from 114 structured interviews to coffee shop owners, with a set of questions in a standardized order. Interviews were conducted from May 2018 to July 2019,

and in early 2020 across 15 cities in nine states in Mexico. A comprehensive directory or list server of specialty coffee shops in Mexico does not exist. We used the list of 46 coffee shops registered with the Mexican Association of Specialty Coffee Shops (AMCCE) as of 2018. All coffee shop owners on this list were contacted, and those who accepted were interviewed. Other coffee shops were identified and refereed by those individuals interviewed (i.e., a snowball technique was also implemented). A total of 106 (93%) owners were interviewed in person and 8 (7%) were interviewed by phone.

### 3.2. Models

We estimated logistic regressions to explore the determinants for coffee shop buying specialty coffee with the variables discussed above. The dependent variable is a binary variable, which equals one if the coffee shop owner/buyer buys specialty coffee and zero otherwise.

For this study we consider that a logit model is appropriate since the buyer is faced with a binary decision to buy or not to buy specialty coffee. All specialty coffee shops buy high-quality coffee, but buyers have the choice to buy high-quality coffee graded as premium coffee (i.e., 80 to 84 cup points) or specialty graded coffee (i.e., $\geq 85$ points), the highest quality grade according to the SCAA standards. Thus, underlying the observed dichotomous response for whether a coffee shop $i$ located in the city $j$ buys specialty coffee or not $(y_{ij})$, there is an unobserved continuous response representing the propensity to buy specialty coffee $(y_{ij}^*)$.

Thus, the propensity to buy specialty coffee $(y_{ij})$ model for a coffee shop $i$ located in the city $j$ is specified as:

$$y_{ij}^* = \beta_0 + \beta_1 x_{2ij} + \ldots . \beta_p x_{pij} + \xi_{ij} \tag{1}$$

such that:

$y_{ij} = 1$ if $y_{ij}^* > 0$, and
$y_{ji} = 0$ if $y_i^* \leq 0$.

where $y_{ij}$ is a binary variable indicating whether the coffee shop buys specialty coffee or not (1, 0 values); and $x_{ij}$ represents vectors of explanatory variables affecting the propensity to buy specialty coffee quality. The coefficients, $\beta$'s, are the parameters to be estimated, and $\xi_{ij}$ is an error term, which assumes a normal distribution $(0, \tau)$ and is independent of the $x$.

It might be the case that the purchasing behavior of coffee shops located in the same city $j$ are independent given the observed covariates, or in other words, that the residuals of the coffee shops that belong to the same cluster are independent. If this were the case, then the simple logit model, Equation (1), would suffice to model purchasing behavior.

Alternatively, it might be the case that coffee shops operating in the same city have similar unobserved characteristics such as entrepreneurial ability, organizational culture, or technology adoption strategies. Thus, to capture unobserved characteristics in clusters, we also consider a multilevel logistic regression model for clustered dichotomous responses [60]. For the multilevel logistic regression, we split the total residual into a two-error component:

$$\xi_{ij} = \zeta_j + \varepsilon_{ij} \tag{2}$$

Substituting $\xi_{ij}$ into Equation (1), we obtain a linear-random-intercept model with covariates:

$$y_{ij}^* = \beta_0 + \beta_1 x_{2ij} + \ldots . \beta_p x_{pij} + \zeta_j + \varepsilon_{ij} \tag{3}$$

$$y_{ij}^* = (\beta_0 + \zeta_j) + \beta_1 x_{2ij} + \ldots . \beta_p x_{pij} + \varepsilon_{ij} \tag{4}$$

Equation (4) is a regression model with the city-specific intercept $\beta_0 + \zeta_j$. The random intercept component $\zeta_j$ is a random parameter that is not estimated along with the fixed

parameters $\beta_0$ through $\beta_p$, but whose variance $\phi$ is estimated together with the variance $\theta$ of the $\varepsilon_{1j}$.

## 4. Results

### 4.1. Descriptive Statistics

Of the total number of coffee shop owners interviewed, 46 (40%) were located in Mexico City, the capital of the country, 31 in Veracruz (27%), 18 in Oaxaca (16%), 11 in Puebla (10%), and a smaller proportion of three coffee shops or less are in the states of Guanajuato, Jalisco, Michoacán, Nuevo León, and Tamaulipas (Table 2).

**Table 2.** Sample of coffee shops in Mexico.

| State | Annual GDP Per Capita (MXN Pesos) [1] | City or Town | Coffee Shops Sampled (n) | Shares (%) | Coffee Shop Age (Years) |
|---|---|---|---|---|---|
| Mexico City | 3465 | Ciudad de México | 46 | 40.35 | 4.1 |
| Guanajuato | 2480 | Guanajuato | 3 | 2.63 | 1.7 |
| Jalisco | 3145 | Guadalajara | 2 | 1.75 | 10.3 |
| Michoacán | 2377 | Morelia | 1 | 0.88 | 25.0 |
| Nuevo León | 3771 | San Pedro Garza | 1 | 0.88 | 0.3 |
| Oaxaca | 1457 | Oaxaca | 18 | 15.79 | 5.3 |
| Puebla | | Puebla | 9 | 7.89 | 5.7 |
| | 2139 | San Andrés Cholula | 2 | 1.75 | 4.0 |
| Tamaulipas | 2845 | Tampico | 1 | 0.88 | 1.0 |
| | | Xalapa | 8 | 7.02 | 1.6 |
| | | Coatepec | 7 | 6.14 | 5.4 |
| | | Huatusco | 5 | 4.39 | 9.1 |
| | | Veracruz | 4 | 3.51 | 3.0 |
| Veracruz | 1792 | Córdoba | 3 | 2.63 | 3.0 |
| | | Boca del Río | 1 | 0.88 | 10.0 |
| | | Coscomatepec | 1 | 0.88 | 9.0 |
| | | Orizaba | 1 | 0.88 | 2.0 |
| | | Cuitláhuac | 1 | 0.88 | 3.0 |
| Total and average years | | | 114 | 100.0 | 4.8 |

Source: Own elaboration. [1] GDP with data from [61].

Table 3 shows descriptive statistics of the variables in the empirical models. Variance inflation factors (VIF) and a correlation matrix are calculated to assess the potential of multicollinearity (Tables S3 and S4 in the Supplementary File). It is unlikely that multicollinearity affects the precision of the parameters' standard errors or parameter magnitudes given that the VIF values are less than or equal to 10.

**Table 3.** Descriptive statistics of variables in the empirical models.

| Variable Categories | Variable | Variable Type/Unit | Shops with Available Data | Average Value | Standard Deviation | Minimum | Maximum |
|---|---|---|---|---|---|---|---|
| $Y$ (Independent variable) | 1 if coffee shop buys specialty, 0 else | Dummy | 112 | 0.38 | 0.49 | 0 | 1 |
| Material attributes | *Professional* | Dummy | 114 | 0.89 | 0.32 | 0 | 1 |
| | *Roast* | Dummy | 113 | 0.27 | 0.44 | 0 | 1 |
| | *Aroma* | Dummy | 114 | 0.31 | 0.46 | 0 | 1 |
| | *Altitude* | Dummy | 113 | 0.33 | 0.47 | 0 | 1 |

**Table 3.** *Cont.*

| Variable Categories | Variable | Variable Type/Unit | Shops with Available Data | Average Value | Standard Deviation | Minimum | Maximum |
|---|---|---|---|---|---|---|---|
| Symbolic attributes | *History* | Dummy | 113 | 0.09 | 0.29 | 0 | 1 |
| | *Value-Added* | Dummy | 112 | 0.11 | 0.31 | 0 | 1 |
| | *Oaxaca* | Dummy | 114 | 0.60 | 0.49 | 0 | 1 |
| | *Guerrero* | Dummy | 114 | 0.21 | 0.41 | 0 | 1 |
| Coffee shop characteristics | *Age_Shop* | Years | 114 | 4.78 | 5.62 | 0.08 | 25 |
| | *Visibility* | Dummy | 113 | 0.95 | 0.23 | 0 | 1 |
| | *W_hours* | Hrs./week | 113 | 78.64 | 15.32 | 35 | 119 |
| | *Ground* | Dummy | 114 | 0.06 | 0.24 | 0 | 1 |
| | *Intermediary* | Dummy | 114 | 0.37 | 0.48 | 0 | 1 |
| | *Young_Cust* | Dummy | 111 | 0.63 | 0.48 | 0 | 1 |
| Coffee shop owner's profile | *Age_Own* | Years | 114 | 35.46 | 8.86 | 22 | 71 |
| | *Gender_Own* | Dummy | 113 | 0.30 | 0.46 | 0 | 1 |
| | *Education_Own* | Dummy | 113 | 0.70 | 0.46 | 0 | 1 |
| | *Knowledge* | Dummy | 114 | 0.11 | 0.31 | 0 | 1 |
| | *Unique_Owner* | Dummy | 114 | 0.46 | 0.50 | 0 | 1 |
| Socio-economic characteristics | *Population* | Ln (inhabitants) | 114 | 12.55 | 1.10 | 9.21 | 14.22 |
| | *Education* | Composite index | 114 | 0.73 | 0.09 | 0.41 | 0.87 |

Variable definitions in Table 1. Italics indicate name of variables as used in the regressions.

First, 89% of coffee bought by coffee shops was tasted by coffee professionals before purchase. This high proportion is common in coffee shops because is in the best interest of coffee shops to ensure high-quality products. In addition, 27%, 31%, and 33% of the coffee stores have roast, aroma, and altitude as the main attributes to be considered for purchasing specialty coffee.

On average, coffee shops have operated 4.8 years, with a minimum of 0.3 and a maximum of 25 years. Most coffee shops are in visible locations (95%), and they are open to the public 78.6 h per week. Only 6% of the coffee shops sell ground coffee, and 37% are intermediaries. Most coffee shop customers are young, with 63% of coffee shop owners indicating to have consumers between 21 and 30 years old mainly.

The average age of coffee shop owners is 35 years old, with 70% of them being female. 70% of the coffee shop owners have a bachelor's or higher degree, and 11% know how to roast coffee. 46% of coffee shop owners interviewed are unique owners.

Finally, 9% of coffee shops in our sample have the history of the coffee as the main symbolic attribute driving the business, and 11% have the added-value concept of coffee as the main business model. A total of 60% of the coffee in these shops originates from the state of Oaxaca and 21% from the state of Guerrero, two relevant coffee regions in Mexico.

*4.2. Estimated Parameters*

Six models are estimated using Equations (1) and (4), with results showing estimated coefficients in Table 4—to highlight the signs of estimated parameters—and odd ratios in Table 5—to highlight the impact of estimates. Models were estimated considering material attributes, symbolic attributes, and coffee shop characteristics (m1 and m4 columns, in Table 4), adding coffee shop owner's profile (m2 and m5), and socio-economic variables (m3 and m6). We included extended versions of Tables 4 and 5 (Tables S1 and S2) in the Supplementary File.

**Table 4.** Models on the probability of purchasing high-quality coffee in coffee shops in Mexico (parameter estimates).

| Variable Name | m1 | m2 | m3 | m4 | m5 | m6 |
|---|---|---|---|---|---|---|
| *Material attributes* | | | | | | |
| Professional | −3.624 *** | −4.083 *** | −5.147 *** | −3.591 *** | −4.126 *** | −4.564 *** |
| Roast | −2.168 ** | −2.45 4** | −2.094 * | −2.099 ** | −2.284 ** | −1.837 |
| Aroma | 2.487 ** | 2.565 * | 2.295 | 2.377 * | 2.463 * | 2.211 |
| Altitude | −1.236 | −1.747 | −1.79 | −1.073 | −1.589 | −1.412 |
| *Symbolic attributes* | | | | | | |
| History | −0.279 | −0.512 | −0.942 | −0.135 | −0.417 | −0.896 |
| Value-Added | 2.208 ** | 2.363 ** | 2.637 *** | 1.795 * | 1.822 * | 1.957 * |
| Oaxaca | 1.579 ** | 1.825 ** | 2.235 *** | 1.626 ** | 1.981 *** | 2.032 ** |
| Guerrero | −0.681 | −0.329 | −0.728 | −0.498 | −0.0775 | −0.522 |
| *Coffee shop characteristics* | | | | | | |
| Age_Shop | −5.047 | −7.401 | −5.992 | −3.59 | −6.52 | −5.722 |
| Visibility | 1.076 | 1.938 * | 1.322 | 0.812 | 1.496 | 1.095 |
| W_Hours | 0.013 | −0.00514 | −0.00882 | 0.0093 | −0.0119 | −0.0113 |
| Ground | 2.275 ** | 2.990 ** | 2.071 * | 1.748 | 2.171 | 1.78 |
| Intermediary | 0.902 | 0.57 | 1.01 | 0.827 | 0.572 | 0.858 |
| Young_Cust | −0.283 | −0.323 | −0.305 | −0.291 | −0.337 | −0.495 |
| *Coffee shop owner's profile* | | | | | | |
| Age_Own | | 0.0411 | 0.0385 | | 0.0468 | 0.0447 |
| Gender_Own | | −0.495 | −0.609 | | −0.434 | −0.551 |
| Educ_Own | | −0.759 | −0.882 | | −0.856 | −0.889 |
| Knowledge | | 1.721 * | 1.876 ** | | 1.777 * | 1.919 * |
| Unique_Own | | 0.6 | 0.321 | | 0.624 | 0.455 |
| Constant | −0.128 | −0.236 | 4.05 | 0.267 | 0.356 | −0.0855 |
| *Socio-economic characteristics of the context* | | | | | | |
| Population | | | −0.867 ** | | | −0.594 |
| Education | | | 11.42 *** | | | 11.92 ** |
| Key statistics | | | | | | |
| N | 107 | 106 | 106 | 102 | 101 | 101 |
| N_Cluster | | | | 16 | 16 | 16 |
| R2_p | 0.243 | 0.291 | 0.337 | | | |
| Log Likelihood | −54.24 | −50.47 | −47.17 | −52.45 | −48.25 | −45.58 |
| Wald chi2 | 30.84 *** | 35.29 *** | 40.59 *** | 19.91 | 20.54 | 22.49 |
| AIC | 138.5 | 140.9 | 138.3 | 134.9 | 136.5 | 135.2 |
| BIC | 178.6 | 194.2 | 196.9 | 174.3 | 188.8 | 192.7 |
| ROC | 0.813 | 0.834 | 0.861 | 0.806 | 0.826 | 0.849 |
| LR Test | | | | 0.00 | 0.00 | 0.00 |

**Notes:** * $p < 0.10$, ** $p < 0.05$, *** $p < 0.01$. Variables defined in Table 1. Italics indicate name of group variables as used in the regressions. Models m1 to m3 are estimated with Equation (1) and m4 to m6 with Equation (4). Exponentiated coefficients; $R^2$_p is pseudo R-square; AIC is Akaike information criterion; BIC is Bayesian information criterion (BIC); N_Cluster is the number of clusters (cities); Likelihood ratio test (LR test) compares the simple logit model ($H_0$) with the mixed logit model ($H_1$). ROC curve is a plot of the true positive rate (sensitivity) against the false positive rate (1—specificity) for all possible classification thresholds of a diagnostic test. ROC shows the trade-off between sensitivity and specificity (any increase in sensitivity is accompanied by a decrease in specificity). The ROC curves conventionally lie above the diagonal, such that the area under the ROC curve should be greater than 50%. A guide for classifying the accuracy of a diagnostic test is as follows: 0.90–1 = excellent; 0.80–0.90 = good; 0.70–0.80 = fair; 0.60–0.70 = poor; and 0.50–0.60 = fail.

**Table 5.** Models on the probability of purchasing specialty quality coffee by coffee shops in Mexico (odds ratios).

| | Variable Name | m1 | m2 | m3 | m4 | m5 | m6 |
|---|---|---|---|---|---|---|---|
| Material attributes | *Professional* | 0.0267 *** | 0.0169 *** | 0.00582 *** | 0.0276 *** | 0.0161 *** | 0.0104 *** |
| | *Roast* | 0.114 ** | 0.0860 ** | 0.123 * | 0.123 ** | 0.102 ** | 0.159 |
| | *Aroma* | 12.03 ** | 13.00 * | 9.928 | 10.77 * | 11.75 * | 9.131 |
| | *Altitude* | 0.291 | 0.174 | 0.167 | 0.342 | 0.204 | 0.244 |

**Table 5.** *Cont.*

| | Variable Name | m1 | m2 | m3 | m4 | m5 | m6 |
|---|---|---|---|---|---|---|---|
| Symbolic attributes | *History* | 0.757 | 0.599 | 0.390 | 0.874 | 0.659 | 0.408 |
| | *Value-Added* | 9.095 ** | 10.62 ** | 13.97 *** | 6.020 * | 6.187 * | 7.081 * |
| | *Oaxaca* | 4.852 ** | 6.203 ** | 9.347 *** | 5.082 ** | 7.251 *** | 7.633 ** |
| | *Guerrero* | 0.506 | 0.719 | 0.483 | 0.608 | 0.925 | 0.593 |
| Coffee shop characteristics | *Age_Shop* | 0.00643 | 0.000611 | 0.00250 | 0.0276 | 0.00147 | 0.00326 |
| | *Visibility* | 2.932 | 6.946 * | 3.752 | 2.252 | 4.465 | 2.991 |
| | *W_hours* | 1.013 | 0.995 | 0.991 | 1.009 | 0.988 | 0.989 |
| | *Ground* | 9.730 ** | 19.88 ** | 7.932 * | 5.741 | 8.772 | 5.928 |
| | *Intermediary* | 2.465 | 1.768 | 2.747 | 2.287 | 1.771 | 2.358 |
| | *Young_Cust* | 0.754 | 0.724 | 0.737 | 0.747 | 0.714 | 0.609 |
| Coffee shop owner's profile | *Age_Own* | | 1.042 | 1.039 | | 1.048 | 1.046 |
| | *Gender-Own* | | 0.610 | 0.544 | | 0.648 | 0.576 |
| | *Educ_Own* | | 0.468 | 0.414 | | 0.425 | 0.411 |
| | *Knowledge* | | 5.592 * | 6.528 ** | | 5.912 * | 6.815 * |
| | *Unique-Owner* | | 1.821 | 1.379 | | 1.866 | 1.577 |
| Socio-economic characteristics of the context | *Population* | | | 0.420 ** | | | 0.552 |
| | *Education* | | | 91,239.5 *** | | | 150,032.0 ** |
| | Constant | 0.879 | 0.790 | 57.38 | 1.305 | 1.428 | 0.918 |

**Notes:** * $p < 0.10$, ** $p < 0.05$; *** $p < 0.01$. Variables defined in Table 1. Italics indicate name of variables as used in the regressions. Models m1 to m3 are estimated with Equation (1) and m4 to m6 with Equation (4). Exponentiated coefficients.

We estimated simple logit (m1 to m3) and multilevel logit models (m4 to m6). The latter models are suitable where random effects are useful for modeling intra-cluster correlation; that is, observations in the same cluster are correlated because they share common cluster-level random effects. Results in Table 4 show that observations in a same city (cluster) are not correlated according to the LR test. That is, coffee shops in a same city are not associated each other or do not have factors (observables and unobservable) that explain data variability. Then, in our empirical multilevel models (m4 to m6) there are not unobservable variables at city level that affect the estimated parameters. In other words, relative to the multilevel models, the simple logit models m1 to m3 explain better the purchasing behavior of specialty coffee by coffee shops in Mexico. Results of models m4 to m6 are shown in Table 4 for completeness.

In m1, material and symbolic attributes as well as coffee shop characteristics are jointly significant, according to the Wald chi$^2$ test in Table 4. Similarly, the simple logit models are significant at the 1% level when owner's characteristics variables (m2) and socio-economic attributes (m3) are added. Simultaneously, m1, m2, and m3 have similar Akaike information criterion (AIC) values of around 138 to 141. However, the Bayesian information criterion (BIC) performs notably better in m1 than in the two other models, m2 and m3, with a value of 178.6. Consequently, we select m1 as the best fit model for our main discussion, implying that adding coffee shop owners' profile and socioeconomic variables to the basic model (m1) do not significantly increase the model's explanatory power. In other words, coffee shop owner's profile and socioeconomic variables jointly do not improve the goodness of fit of models m2 and m3 for purchasing probability of specialty coffee. A caveat to the previous statement: models m2 and m3 are statistically significant and with signs of estimates consistent with m1, meaning that owner characteristics and socio-economic conditions might be relevant for the research question in this study, it is just that the owner characteristics and the socio-economic variables jointly are not significant and therefore do not contribute to explain the total variability of the model.

Among material quality attributes, in m1, we found that roast negatively influences the purchase probability of specialty coffee, at 5% significance level. Similarly, aroma is positive and statistically significant at 5%. The variable altitude is not significant in any

model. Of particular interest is the negative and statistically significant sign of professional. Additional analysis for this estimate is provided in the next section.

From the set of symbolic attributes, estimate Oaxaca is positive and statistically significant at 5% level, indicating that having available coffee produced in the State of Oaxaca increases the likelihood of coffee shops buying specialty coffee. With regards to the coffee shop's core business model concept, the value-added business model is also a positive and statistically significant attribute determining the probability of purchasing specialty coffee by coffee shops. Regarding coffee shops characteristics, only the variable ground is statistically significant at 5% level. This means that coffee shops selling ground coffee rather than only in-store drinks, are more likely to buy specialty coffee.

*4.3. Odd Ratios*

Table 5 provides the odd ratios. An extended version of this table is included in the Supplementary File. Keeping other independent variables constant, the odds of purchasing specialty coffee by coffee shops are about eight times as low (1/0.114; Table 4) for shops considering roast as the main characteristic to be purchased. In contrast, the odds of purchasing specialty coffee are about 12 times higher for coffee with aroma as the main characteristic driving purchasing decision.

The probability of purchasing specialty coffee increases about five times if the coffee offered is produced in the State of Oaxaca, relative to coffee from other states (Table 5). Regarding value-added, this variable increases about nine times the probability of buying specialty coffee.

The result regarding the presence of coffee professionals evaluating coffee to be purchase by coffee shops is of particular interest. Since there is a direct relationship between coffee quality bought and price, a coffee professional will reduce operating costs by purchasing either premium quality or conventional blends since the coffee professional may use his/her expertise during the preparation of coffee drinks and meet customers' expectations. According to the results, the odds of purchasing specialty coffee are about 35.5 times (1/0.0267) less for coffee shops with coffee professionals who tasted the coffee before purchase. For an alternative interpretation, we calculated marginal effects in Table 6 for the variable at 0 (e.g., no presence of a coffee professional in the coffee purchasing decision) and 1 level (e.g., presence of a coffee professional).

**Table 6.** Marginal effects for "professional" on the probability of purchasing specialty coffee by shops in Mexico.

| Category | Marginal Effect | Robust Standard Error | Z |
|---|---|---|---|
| No coffee professional | 0.90 *** | 0.072 | 0.000 |
| Coffee professional | 0.34 *** | 0.042 | 0.000 |

Marginal effect estimation for variable *Professional* in model m1, as described in Table 4. *** $p < 0.01$.

When coffee professionals do not taste the coffee before purchase, coffee shop owners' marginal probability to buy high-quality coffee is 90%. In contrast, coffee shops' marginal probability of buying high-quality coffee is 34% when coffee professionals tasted the specialty coffee, as shown in Table 6. This confirms the negative relationship between coffee professional tasting coffee before purchase and the probability to buy specialty coffee.

Finally, the probability of purchasing specialty coffee increases about nine times if coffee shops sell ground coffee rather than only in-store drinks (Table 5). This implies that product diversification in the coffee shop and specialty coffee are positively correlated.

## 5. Discussion

*5.1. Linkages and Contrasts between Results and the Existing Literature*

Many have studied the relationship between specialty coffee quality and symbolic attributes [62–65] and only recently specialty coffee quality was modeled as a function of both symbolic and material attributes [18]. This literature evaluated coffee tasters' and

international coffee buyers' preferences using data from the Cup of Excellence auctions. Unlike previous research, we built a database of coffee shops commercializing high-quality coffee in Mexico. This unique dataset allows us to extend the framework in [18] by including as vectors of explanatory variables: (a) material attributes, (b) symbolic attributes, (c) coffee shop characteristics, (d) profile of the coffee shop's owner, and (e) socio-economic variables of the cities where the coffee shops were located.

Broadly, our results are consistent with the expectations discussed in the hypotheses section and summarized in Table 1. That is, the likelihood of purchasing specialty coffee increases when: coffee's aroma drives the purchase, coffee purchased is from the state of Oaxaca, the coffee shop has a value-added business model, the coffee shop is diversified selling both ground coffee and coffee drinks, the coffee shop owner's knowledge on coffee supply chain activities is high, and the coffee shop is located on a city with a higher education index. In contrast, the likelihood of purchasing specialty coffee decreases when a coffee professional tastes the coffee before the purchase, when coffee shops are not given the opportunity to roast their own coffee (e.g., they roast they own coffee), and in coffee shops located in larger cities.

These findings are relevant for coffee shops (e.g., demand) and coffee farmers or processors/roasters (e.g., supply). From the perspective of coffee bean suppliers, one implication is the recognition that coffee shops selectively consider material quality attributes as part of the buying decision process, which is consistent with previous research [12]. For instance, even though altitude is widely recognized to affect coffee's quality [5,28,34,35], coffee shops in Mexico consider coffee's aroma as the most appreciated material attribute when buying high-quality coffee and altitude does not affect their purchase decision. This suggests that specialty coffee shops target coffee producers from high elevation coffee regions only. Coffee's aroma has been extensively documented as a highly valued attribute from the perspective of the final consumer even at the retail level [35,37–41]. Our work is the first to document the value of this attribute for the coffee shop buyer. This is not trivial since quality definitions are not necessarily consistent among coffee supply stakeholders [12,66], further discussed in the next section.

Another implication for specialty coffee suppliers is the recognition that coffee produced in the state of Oaxaca is highly valued by coffee shops in Mexico. Coffee stakeholders should recognize that recent promotion efforts by farmers and institutions throughout the Oaxaca Coffee Quality Award (explained in the hypotheses section) might have played a role on coffee shops preferring coffee from Oaxaca. Other producing regions might follow such efforts, as is indeed happening in Mexico (e.g., the state of Guerrero). This finding is consistent with current efforts in the state of Oaxaca to obtain denomination of origin status [67], which in turn has the potential to provide marketing advantages for farmers [68].

On the buyer side (e.g., the coffee shop), our analysis implies that more appealing/higher quality coffee does not necessarily translate into high-grade coffee drinks; and that the participation of coffee professionals in the buying decision process is relevant for coffee shops. This finding is consistent with previous research that professional coffee tasters drive market values in specialty coffee markets [12]. Analyzing whether enough coffee professionals are available in the coffee shop Mexican segment and supporting training of additional professionals, if needed, might directly benefit the specialty coffee segment and indirectly benefit the overall coffee supply chain.

The previous implications illustrate only instances—we do not discuss each finding for the sake of brevity—of how results of this research could be useful for coffee supply chain participants. In a broader sense, our research results relate the understanding of quality and efforts by agricultural enterprises transitioning from a supply chain approach to a value-based supply chain (VBSC) approach.

*5.2. Major Themes and Limitations*

Findings of this research relate to two major themes: the understanding of quality across specialty coffee stakeholders and the integration of specialty coffee VBSCs. There are divergences on the understanding/definition of quality across coffee value chain stakeholders [12,66]. This is not surprising since quality of a coffee drink is a multifactorial and complex trait, with agricultural practices influencing 40% cup quality attributes and postharvest processing technology affecting 60% quality attributes [69]. In addition, each supply chain stakeholder understands quality in terms of how that chain player contributes to overall product quality. For instance, the coffee farmer's construct of coffee quality focuses on material attributes related to coffee production of green coffee bean while a roaster and a coffee shop may focus on material attributes at processing and on symbolic attributes. Recent analysis with Colombian coffee farmers and roasters in Vienna [66] and meta-analysis on market value of coffee and functional biodiversity [12] highlight the divergences on the understanding of specialty coffee quality.

The coffee literature calls for a better understanding of how coffee quality definitions can be more consistently recognized across stakeholders [12]. Our study contributes a little in this regard by being the first to identify the quality attributes that are of upmost importance to specialty coffee shops, and reflects on how this identification may impact smallholder coffee farmers. One limitation of our proposed framework is the lack of in-person attributes measurements [5,18]. In-person attributes result from human interaction between baristas, for instance, and end consumers at specialized coffee shops at the moment of consumption.

A transition from a supply chain approach to a VBSC approach might contribute to unify the understanding of quality and help smallholder coffee farmers in several ways. Material and symbolic quality attributes are generally irrelevant to coffee farmers unless they are involved in roasting, cup tasting, or directly in specialty coffee markets [12]. The involvement of coffee farmers with roasters, coffee shops or other stakeholders has been termed in the specialty coffee literature as "direct relationship coffee", defined as transactions based on a direct producer-roaster relationship guided by personal visits, transparency, trust, among other dimensions [66,70]. More generally, direct relationship coffee is embedded in the VBSC framework, the latter defined as an industry mode where cooperation between farmers and other stakeholders including processors, retailers, and consumer exists [71,72]. VBSCs are challenged to organize but have the potential to achieve higher performance and equitable profits across partners.

Our research suggests that the specialty coffee niche in Mexico has some elements required to integrate a VBSC. There is anecdotal evidence that some coffee shops in our sample maintain a direct or semi-direct (through roasters) relationship with smallholder coffee farmers and that there is information sharing among trading parts. For instance, some coffee shop owners indicated that they support smallholder farmers with technical assistance in coffee plantations and prefer to purchase specialty coffee from roasters supporting local smallholder producers. Several coffee shops owners in this sample indicated that they travel once or twice a year to the coffee-growing regions that supply their coffees to learn more about the production and transformation process of the coffees they buy and sell. A few coffee shops have even extended their relationship with their suppliers by providing support in health services for the smallholder farmers. A successful integration of specialty coffee VBSCs in Mexico has the potential to make smallholder coffee farming more economically sustainable. The price that coffee producers receive for selling specialty coffee quality to coffee shops is relatively high compared to local and international buyers in conventional marketing channels [13,14]. VBSCs improve information sharing which in turn could facilitate the understanding of quality definitions/measures across the supply chain as well. Further, the direct contact between coffee growers/sellers and participants of the coffee supply chain is associated with sustainable production practices such as higher tree diversity, saving water practices, higher use of organic agronomic practices, among others [10–12].

## 6. Conclusions

We built and analyzed a database of coffee shops commercializing high-quality coffee across nine states in Mexico. This unique database allowed us to model the purchase of specialty coffee as a function of: (a) material attributes, (b) symbolic attributes, (c) coffee shop characteristics, (d) profile of the coffee shop's owner, and (e) socio-economic variables of the cities where the coffee shops are located. Our models extend the framework used in prior research. Overall, our results are consistent with expectations developed from the coffee literature. That is, the likelihood of purchasing specialty coffee increases when: coffee's aroma drives the purchase, coffee purchased is from the state of Oaxaca, the coffee shop has a value-added business model, the coffee shop is diversified selling both ground coffee and coffee drinks, the coffee shop owner's knowledge on coffee supply chain activities is high, and the coffee shop located in a city with a higher education index. In contrast, the likelihood of purchasing specialty coffee decreases when a coffee professional tastes the coffee before the purchase, when coffee shops are not given the opportunity to roast their own coffee, and in coffee shops located in larger cities.

Our findings are relevant to both coffee suppliers and coffee buyers, and we discuss some implications of our findings for both the supply and demand sides. More generally, we discuss how our findings may consolidate understanding of coffee quality across specialty stakeholders. Our research suggests the specialty coffee niche in Mexico has some elements required to support value-based supply chains. In this sense, our findings are relevant for governance and policy-making [73]. Despite efforts by the Mexican government to support smallholder coffee farmers, improvements are needed in the articulation of sustainable value networks in the coffee business, which could be achieved through VBSCs. Research results could be extended to other coffee producing regions worldwide.

**Supplementary Materials:** The following are available online at https://www.mdpi.com/article/10.3390/su13073804/s1, Table S1: Models on the probability of purchasing high-quality coffee in coffee shops in Mexico (parameter estimate and standard errors), Table S2: Models on the probability of purchasing specialty quality coffee by coffee shops in Mexico (odds ratios and standard errors), Table S3: Variance Inflation Factor, Table S4: Correlation matrix.

**Author Contributions:** Conceptualization, R.S.-J. and C.J.O.T.-P.; methodology, R.S.-J. and Á.R.-D.; investigation, R.S.-J. and A.Y.P.-V.; data curation, R.S.-J.; writing—original draft preparation, R.S.-J., Á.R.-D. and C.J.O.T.-P.; writing—review and editing, R.S.-J., Á.R.-D. and C.J.O.T.-P.; funding acquisition, R.S.-J. and C.J.O.T.-P. All authors have read and agreed to the published version of the manuscript.

**Funding:** This research was funded by the National Council of Science and Technology (CONACYT) in Mexico, Scholarship number #611435, the Sectorial Fund SAGARPA-CONACYT (grant number 2016-2101-277838), Colegio de Postgraduados Campus Cordoba in Mexico, and The University of Tennessee, Institute of Agriculture in United States. Trejo-Pech acknowledges that this work was partially supported by the United States Department of Agriculture's National Institute of Food and Agriculture, Hatch Multi-State project 1020537.

**Institutional Review Board Statement:** The study did not require ethical approval.

**Informed Consent Statement:** Informed consent was obtained from all subjects involved in the study.

**Data Availability Statement:** The data presented in this study are proprietary information from Colegio de Postgraduados (COLPOS)/National Council of Science and Technology (CONACYT) and is available on request from the corresponding author.

**Acknowledgments:** Servín-Juarez acknowledges support provided for this research during her Sabbatical Year at The University of Tennessee, Institute of Agriculture in United States; The National Council of Science and Technology (CONACYT), Colegio de Postgraduados Campus Cordoba and all coffee shop owners, key informants, enumerators, and the research team in Mexico for all their valuable support to conduct the investigation.

**Conflicts of Interest:** The authors declare no conflict of interest.

## Appendix A

**Table A1.** Variables and selected authors in the literature review.

| | Variable | Selected Authors |
|---|---|---|
| Material attributes | Professional | [10,19,34–36,55,74–84] |
| | Roast | [34,35,77,80] |
| | Aroma | [35,36,38–41,76,80,85,86] |
| | Altitude | [5,30,42,43] |
| Symbolic attributes | History | [87,88] |
| | Value-Added | [14,44,45] |
| | Oaxaca | [45,46] |
| | Guerrero | [46] |
| Coffee shop characteristics | Age_Shop | [47] |
| | Visibility | [48,50,89] |
| | W_Hours | [35,80,85,90] |
| | Ground | [7,14] |
| | Intermediary | [45,51] |
| | Young_Cust | [35,36,91] |
| Coffee shop owner's profile | Age_Own | [35,36,92,93] |
| | Gender_Own | [93,94] |
| | Educ_Own | [34,52,53,95] |
| | Knowledge | [7] |
| | Unique_Own | [9,52–54,95–98] |
| Socio-economic characteristics of the context | Population | [14] |
| | Education | [35,57,58,99,100] |

Source: Own elaboration.

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
