# Peer review of "Specialty Coffee Shops in Mexico: Factors Influencing the Likelihood of Purchasing High-Quality Coffee"

_sustainability, doi:10.3390/su13073804_

Round 1

Reviewer 1 Report

This manuscript would require less modifications if submitted to a journal such as MDPI Agriculture since it focuses on the economics of factor markets for Mexican specialty coffee (agriculture). In order to make this manuscript focused on Sustainability (economic, environment, community), there needs to be additional writing and citations of research to explain how the research and research results adds to the body of knowledge on such sustainability. Please make the following SEVEN substantive changes described below and line number specific edits. This reviewer would be open to re-evaluating the manuscript after these changes are made:

1) In the abstract, keywords, introductory sections, and discussion the authors need to make it much more clear how the econometric research results and implications of these research results make the area more sustainable. I am assuming from the context of the article that the preferences of these coffee shops owners are representative, then this suggests that smaller rural shareholder farmers would be more able to meet the unique demands of these shops. This could support local producers and local economies relative to more industrialized coffee plantations. How can you quantify these positive community benefits or at least devote a significant amount of the discussion to explaining these benefits. If the smaller coffee producers have more sustainable practices (e.g. shade coffee and more ecologically diverse farm landscapes) then this suggests there are positive environmental benefits as well. How can these be quantified in your research or at least substantively discussed in the Discussion section? Otherwise, the manuscript focuses exclusively on economic factor markets which by definition is not sustainability (economics, environment, community) and merely only one facet of it. Another alternative would be to submit to MDPI Agriculture which would not require as much revision.

2) The manuscript needs to have a part of the introductory section devoted to explaining 1) above and citing literature. Please make room for this by decreasing the length of the current section 2. Consolidate this into one section Background and break out into sub-sections accordingly. The length of this new writing should be about as long as the current section 2. Rename the current section 3 as 3. Hypotheses.

3) Sections 4 and 5 are for the most part Methods so consolidate into a new section Methods and break out into sub-sections accordingly.

4) The current section 6 should be split into separate sections labeled as Results and 6. Discussion

5) The current section 7 should be Conclusion with any writing on implications moved to the new discussion section and the conclusions involving bigger picture summary of the research conducted with respect to what the authors are concluding

6) Summarize hypotheses in L157-384 into Table 1 in Hypotheses (this is currently Table A1 in the Appendix so please move this) and then consolidate the writing to point out general expectations of results. This could cut the size of this section down to ¼ or so of the length which allows more room for the new section described in 2) above. Change this table the way it currently is to add a new column to describe the variable. Instead of doing the last column as “Selected Authors” change this to describing the geographic region for the crop (e.g. coffee) and then keep the citations as [#] as this will save a lot of space. The bulk of the table emphasizes the author names of the cited research which is not needed since one can go to the References section to find this out.

7) Please as briefly as possible add to Methods and 5. Results regarding how you tested for multicollinearity to rule out its impacts on biasing parameter estimates in your logistical regression models (e.g. correlation matrix or variance inflation factors which should be put into a table in the Appendix).

Specific Line Number of Manuscript Figure/Table comments:

L2-3 – The title of peer-reviewed journal articles typically do NOT capitalize all words in the title (theses and dissertations typically do this) so please change to: “Specialty coffee shops in Mexico: Factors influencing the likelihood of purchasing high-quality coffee”

L13 – Change to “shop owners across 15 cities in nine states in Mexico.”

L20 – Value-added? What exactly do you mean by added value if this is not value-added (adding value such as roasting coffee is adding value to raw coffee beans)

L23 – What does Q stand for? Quality?

L46 – Just state as 9% or ~9%

L51 – What about the community and environment as well? Need to cite more sources on that?

L68 – How EXACTLY contribute to sustainability? Please be more specific

L69 – No need to state authors of cited research so just use [#] to save space

L87 – Use e.g. and not i.e. as this is an example

L91 – No need to state authors of cited research so just use [#] to save space

L97 – Change to “between producers and coffee shops in factor rather than product markets.”

L115 – No need to state authors of cited research so just use [#] to save space

L120 – No need to state authors of cited research so just use [#] to save space

L127 – No need to state authors of cited research so just use [#] to save space

L147 – Change to “…early 1990s as they are distinguished…”

L157-384 – This entire section uses A LOT of writing to go step by step in describing hypotheses of the logistical model so while fine in a master’s thesis or PhD dissertation, for a journal article this needs to be summarized more and reduced in length (in other words what are the major points in your expected results that you want to convey and then write in a CONCISE way that informs the reader and does not come across as a laundry list of every single variable you used or tested in your econometric models)

L158-159 – No need to state authors of cited research so just use [#] to save space

L176 – Remove “Sensory test.” as there is no need for a sub-section header here

L195-196 – No need to state authors of cited research so just use [#] to save space

L202 – No need to state authors of cited research so just use [#] to save space

L245 & L247 – Write as “US$ 5.83” and “US$ 6.24”

L254 – English tons or metric tons (a.k.a. Mt or t)?

L258-265 – Indent so consistent

L266 – No need to state authors of cited research so just use [#] to save space

L364-365 – No need to state authors of cited research so just use [#] to save space

L386-390 – This belongs in 1. Introduction

L391-396 – This writing and table belongs in 5. Results

L405-413 – This writing and table belongs in 5. Results

L414-445 – Most readers of MDPI Sustainability are not neo-classical economists so you need to explain dummy variables for them as well as make the writing in this part of the manuscript much shorter and less technical

L446-461 – This writing belongs in 5. Results

L462 – For the new 4. Methods section you need to create sub-sections on how you conducted your survey as well as the econometric model

L614 – Use e.g. and not i.e. as this is an example

Author Response

Dear Reviewer,

We appreciate it very much your suggestions for improvement of the paper. I am sending you in attachment a file with the information required in your revision and work we did is specified in the responses and in the updated manuscript. 

Reviewer 2 Report

The authors analyse data collected from 114 in – depth personal interviews with coffee shop owners across 15 cities belonging to nine states of Mexico. They find that the coffee’s aroma is the most appreciated material attribute by coffee shops. They also underline that the participation of coffee professionals in the buying decision process is relevant for coffee shops. The authors give a detailed description of all the factors influencing the purchasing probability of high quality products. The description is well organized and sufficiently clear. All the variables considered are correctly defined and appropriately included in the six model specifications that are tested in the article. The research methodology is suitable and well performed.

I suggest to modify the last section “Implications and Conclusions” describing better the results of empirical analysis and illustrating what effectively the policy makers together with the other stakeholders can do in order to improve the economic performances of the sector.  

Author Response

Dear Reviewer,

We appreciate it very much your suggestions for improvement of the paper. We included new information in the Discussion and Conclusion sections. You can verify the information in the updated manuscript. And we will very happy to attend any further recommendation for improvement of the manuscript. 

Reviewer 3 Report

The paper covers an important topic and relevant conclusions are achieved. The authors studied models the purchasing behavior of high-quality, specialty coffee, by coffee 11 shops in Mexico, and analyzed data collected from 114 in-depth personal interviews with coffee 12 shop owners across 15 cities, in nine states, in Mexico). However, some aspects must be changed before being considered for publication:

  • The keywords can be improved, for example including “high-quality coffee” instead of “small business”;
  • The introduction can be improved, for example emphasizing the novelty and contributions to the literature;
  • Governance and policy making can be a relevant matter for the development this kind of projects (vide Marques and Pinto, 2018);
  • In the end of the introduction, a paragraph presenting the organization of the paper should be added;
  • Try to include a table summarizing main achievements in literature review;
  • All the abbreviations have to be presented in the text and do not repeat the presentation of the abbreviation;
  • Regarding the method, the authors could justify why this model was chosen;
  • Explain the contents of the parameters;
  • More information can be provided about the case study (GDP, GDP per capita, and if possible per region, …);
  • All sources of information must be presented in the text;
  • Present the unit of variables;
  • Explain better the statistical analysis of the variables and its meaning, in particular the sign of the variables;
  • The authors can also provide more explanations about the differences between the results achieved and the expected ones;
  • I would also expect more practical recommendations for decision makers and policy in this regard;
  • The references must be in line with author guidelines. For example, some issues are missing.

References:

MARQUES, R.; PINTO, F. (2018). How to watch the watchmen? The role and measurement of regulatory governance. Utilities Policy. Elsevier. ISSN: 0957-1787.  Vol. 51, pp. 73-81.

Author Response

Dear Reviewer,

We appreciate it very much your suggestions for improvement of the current version of the manuscript. We are sending you in attachment a file with the responses to your questions. We will be very happy to receive any further suggestion.

Reviewer 4 Report

This paper is an interesting read with a topic of coffee shop owners in Mexico.  You well attempted to collect data from 114 in-depth personal interviews with coffee shop owners across 15 cities, in nine states, in Mexico. You should have described how you conducted the in-depth interviews. (When did you conduct the interviews? If it was during the pandemic, you also should explain how to conduct the interviews following the social distancing rule.)

      One critical thing of this research is the methodology and the justification for it. You described that you collected interview data from in-depth interviews, but your dataset seems to be a survey outcome. If this is based on a qualitative approach, where is a justification for your chosen method? If this is a research based on a survey, you also have needed to justify your method. 

You also discussed that you created six models to analyse the participants' data which are summarised in Table 3, but the process and justification for differentiation of the variables (e.g. coffee shop owners' profile, socio-economic characteristics) is not explained fully. 

     Perhaps, the research gap with the rationale for the research should have been discussed in a clearer way, and in so doing, a clearer discussion logic can be presented throughout the paper, which can be valuable for the readers and the field of study. For instance, you discussed '...the identification and understanding of the determinants of specialty coffee purchasing behaviour, the goal of this study, might ultimately benefit coffee producers and contribute to coffee production sustainability', however, in the main text, you did not expand this aim of the research referring to the topics of sustainability, therefore it would not be easy for the journal readers to understand and evaluate the research outcome (for instance, how did you plan to fuse the research aim and the analytical model with specific observed variables demonstrated in Table 3?).

Author Response

Dear Reviewer,

We appreciate it very much your suggestions made to the previous version of the paper. We are sending you a file with the responses to your questions and we will be very happy to incorporate any further suggestion.

Round 2

Reviewer 1 Report

Thank you for making enough modifications to the manuscript to begin to justify why this meets the aims and scope of MDPI Sustainability (economic, environment, community). Please make the following FIVE substantive changes described below and line number specific edits. This reviewer would be open to re-evaluating the manuscript after these changes are made:

1) The additional writing to the manuscript has a lot of sentences that run on and on and are too long. These need to be broken up to make the points each sentence makes so they flow from one sentence to the other. A lot of the writing states that the research you are citing says this or that. Since you are citing via [#] in a lot of these cases you can just state the main point(s) from the literature you are citing and get straight to the point. The additional writing also needs to be cited (e.g. where I have [CITE as #] written in the re-write of the paragraph on L66-86). The authors also need to make the writing more concise and clear. For example, the re-written paragraph on L66-86 (see line specific edits that follow) reduces the word count by 111 words from 357 to 246 and the clarity of the writing is better getting straight to the point and following a logical flow with sentences that are not too long. Please involve an English editor not for grammar so much as paragraph and sentence organization and clarity and conciseness of writing. For example to start a sentence with “In consequence…” is grammatically awkward. The reason that the writing needs to be CLEAR and CONCISE is that most if not all readers mentally shut down if they encounter excessive, redundant, repetitive writing. If you can make things clear with 246 words, why use 357 words?

2) The Hypotheses sections for coffee shop and coffee shop owner characteristics are extremely long and the writing is literally variable by variable over L237-305. While this writing style may be acceptable for a Master’s thesis or PhD dissertation, for an academic journal you risk losing the focus and engagement of the reader since there is too much detail. In this case, having these hypotheses in table format would be clearer than writing in bullet points. Please create a new Table 1 where these and other variables are concisely summarized with expected impact on the dependent variable and then limit the writing describing this. For new Table 1, keep a similar format to Table 2 where the row headers for the independent variable and dependent variables are the same but have the column headers be: 1) “Variable categories”, 2) “Variable”, 3) “Description”, and 4) “Expected correlation with independent variable.” So for the independent variable this would be “n/a” while for all dependent variables you could use “+” or “-” or “+/-” to improve how concise the table can be. When you edit L237-305, the writing should be much more like L155-236 and even the writing in this section should be made more clear and concise (see example of edits to paragraph on L66-86 for style of editing). Please reduce the word count for L155-236 by 33%.

3) The Discussion section as written is Results. You need to put this in the Results section and then do sub-topic headers for Results.

4) The Conclusion section as written is really your Discussion section and needs to be substantively improved (please start with creating outlines for both of the following sections before writing). The Discussion section in most manuscripts involves two sections:

a) Creating contrasts and linkages between your results and results of prior literature. In this case, the research is unique to specialty coffee but have there been similar econometric studies on specialty products in OTHER industries? Here you would explain to the reader the broader relevance of your research results as it compares to prior literature.

b) Discussing major themes or limitations to the implications of your research results. For example, you make a point of saying that these small shareholder supply chains require greater communication and transparency between farmers and coffee shops for example (last paragraph on L525-535). Your research results highlight important significant attributes that shape raw coffee purchases by coffee shops from small shareholder farmers. These are called a “value-based value chains” (VBVCs) in the literature. How are coffee shop owners able to effectively communicate this to farmers from which they purchase coffee? Do they have regular meetings? Is there a shared commitment to support these farmers and share success or do certain input cost minimization requirements need to be met where it is more important to go to the open global market? This does not have to be a discussion point(s) you choose to discuss but right now there is no clear substantive discussion. Set a goal of two substantive discussion points for this.

5) The Conclusion section is typically very brief with 1 to 2 paragraphs. Here you abstract up even more and summarize your results in the context of relevance to sustainability of other industries for example. You also typically end this section covering ways future research could address gaps in understanding that your research did not address. This will need to be written and added.

Specific Line Number of Manuscript Figure/Table comments:

L22 – Change to “On the supply side, coffee growers or”

L27 – Change to “market contributes to the sustainability of coffee supply chains.”

L62 – Change to “…market contributes to the sustainability of coffee supply chains.”

L66-86 – Change so first paragraph is much more concise so it reads “The goal of this study is to identify the factors influencing the purchase of specialty coffee by coffee shops in Mexico. This can benefit more sustainable smallholder coffee farmers selling coffee to coffee shops directly or indirectly through coffee cooperatives by providing them with relevant information to better target their customers. Sustainable small shareholder coffee supply chains can be economically sustained since consumers are willing to pay a higher price for specialty coffee compared to conventional coffee grown on larger industrial plantations [13,14]. These types of coffee supply chains can also be more sustainable from both environmental and community perspectives. For example, smallholder coffee farmers in Colombia work directly with buyers, roasters, and/or importers to achieve desired environmental, socio-economic, technological, and community benefits [11]. Compared to conventional coffee plantations, smallholder coffee supply chains are more associated with sustainable production characteristics such as higher tree diversity for shade-grown systems, improved water and soil conservation, and organic management [CITE as 15]. Smallholder coffee production systems can also empower farmers to better understand and operate within these types of supply chains [CITE as 16]. Recent meta-analyses show smallholder coffee farmers producing certified, high-quality coffee receive more positive than negative impacts across environmental, human, social, natural, and economic dimensions of sustainability [17,18]. Mexico is a relevant coffee producing region supply certified / high-quality coffee markets [19-21] where there are about 500 small coffee farms averaging 1.4 ha/farm which employ more than 3 million workers mostly from poor indigenous communities [19].”

L87-535 – Make sure all [#] are updated if the two [CITE as #] above are NEW citations. If you are using existing citations these updates may be less extensive.

L112-114 – Delete this as the paragraph is redundant. The reader understands this by reading the manuscript.

L382 – For Table 2, please insert a column between the “Variable” and “# Coffee shops with available data” columns and title the new column “Variable Type / Unit” where you then write what the measure is for the variable. For example, for dummy variables you would write “Dummy” and it is clear what this is from the description of Dummy for the independent variable. For age of coffee shop this would be “years.” If you do not do this, then it is not clear if the age of the coffee shop is in months versus years for example.

L388-413 – This footnote is too long. There is no need to repeat every single base definition of a dummy variable for EVERY variable. You can state this in one sentence: “All independent variables in models are also specified as dummy variables.” Also please note that the new Table 1 that you add will make this writing redundant since the new Table 1 defines the sole dependent and large number of independent variables tested in the models with expected sign and impact on the dependent variable.

L537-540 – Please see the instructions for authors for MDPI Sustainability as this section involves initials for all co-authors and the names are not written out in the interest of being space efficient.

Author Response

"Please see the attachment".

Reviewer 4 Report

I would like to appreciate a great effort to revise the manuscript.

Reagrding the conclusion (output of the analyses), I would like to raise two points. Please discuss the points below in details.

(1) Page 7, line 300. You only discussed the 'socio-economic characteristics' in the section 2.5. Socio-economic characteristics, but you did not enhance the relevat discussion in line with the analytical outcomes on page 17, Table 3. In your model 1-3. The impact of socio-economic characteristics of the context plays a critical role in your models, therefore, it would be better for you to enhance the discussion in the context of Socio-economic characteristics in discussion.

(2) It is good to see your discussion on page 12 line 429- '...Then, in our empirical multilevel models (m4 to m6) there are not unobservable variables at city level that affect the estimated parameters. In other words, relative to the multilevel models, the simple logit models m1 to m3 explain better the purchasing behavior of specialty coffee by coffee shops in Mexico.' Can you please discuss why you implemented two different analytical approaches, simple logit models and multilevel models? For instance, in this paper, the term 'multilevel models' appear only three times (page 12, in one paragraph line 424-432), and there are no explanation of the process of building the models based on two different logics.  

Author Response

"Please see the attachment".

Round 3

Reviewer 1 Report

Thank you for making requested edits to the manuscript. Please make these final minor edits before page proof edit stage prior to publication. Specific Line Number of Manuscript Figure/Table comments:

L83 & L86 – Delete both [11] as they are redundant

L192 – Last word in this line should be “in”

L195-196 – All words capitalized should NOT be capitalized

L226-227 – Change to “US$ ?.??/kg (5.83/lb)” and “US$ ?.??/kg (6.24/lb)” since you also need to have these prices listed in US$ per kilogram as well

L233 – Change to “?.?? metric tons (6.39 tons)”

L235 – Remove second redundant period from end of sentence

L303 – In Table 1 you need to reduce the number of words used in the each variable description so it takes up only ONE line and not two which will reduce the size of the table. You can eliminate “Binary =” since you do not classify age for example as “Continuous.” For example:

“1 if specialty coffee purchased, 0 else” and please note that the “Independent” in this row should NOT be italicized to be consistent with the rest of the table

Note that you repeat “purchase decision” for these 4 variables so you can change to “Characteristics influencing or driving coffee purchase decision” and you may have to reduce the width of the first column titled “Variable categories” but you should be able to do this since there is more space to fit in words

“1 if coffee tasted by coffee professional, 0 else”

“1 if coffee roast, 0 else”

“1 if coffee aroma, 0 else”

“1 if altitude of coffee production, 0 else”

Change first column header to “Coffee shop business models and location of coffee purchased”

“1 if coffee history used, 0 else”

“1 if value-added, 0 else”

“1 if purchased from Oaxaca, 0 else”

“1 if purchased from Guerrero, 0 else”

Keep the first column header as written

“Number years coffee shop operating”

“1 if in visible/attractive locations, 0 else”

“Hours open per week”

“1 if sell ground coffee, 0 else”

“1 if shop also is an intermediary, 0 else”

“1 if customers mostly 21-30 years old, 0 else”

Please change first column header to “Coffee shop owner characteristics”

“Age in years”

“1 if male, 0 female”

“1 if bachelor’s or higher degree, 0 else”

“1 if knowledgeable about roasting, 0 else”

“1 if only one owner, 0 if more”

Change first column header to “Socio-economic characteristics of city where coffee shop located”

“Total city population”

“Human development index using literacy & school attendance”

L328 – Both should be “i.e.,” and NOT “e.g.,” since these are not examples but rather the cup points definition of each label

L365 – For Table 2, please put a “,” at thousands place for all MXN pesos values and please round so for example “3,465” and please make sure your titles in all tables are more readable so for example the second column title takes up 3 lines that should be center justified as:

Annual GDP

per capita

(MXN pesos)1

Likewise, the fourth column header title in Table 2 should read:

Coffee shops

sampled (n)

Likewise, the sixth column header title in Table 2 should read:

Coffee shop

age (years)

Also please right justify the numbers in the 5th and 6th columns as the decimals will align for better comparison. Also, the footnotes in Table 2 at the very bottom need to be left justified and not center justified. By creating more width space efficiency you will be able to widen the 1st column and get rid of the hyphens which will allow you to use 1 row rather than 2 rows.

L388 – Please narrow the 2nd column by changing the 2nd row label to “1 if coffee shop buys specialty coffee, 0 else” and the narrowing of the 2nd column should allow you to widen the 5th, 6th, 7th, and 8th columns at the end so the column header labels can actually fit better in 2 rows rather than 3 to 5 rows:

“Avg. value” and “Std. dev.” and “Mini- mum” and “Maxi- mum”

Note that this widening of the last 4 columns should also correct “78.64” etc. taking up 2 rows and also please make sure to right justify the date in these last 4 columns so the decimal points align.

L397-398 – For Table 4, please left justify ALL variable group labels as they are currently center justified and difficult to read. Also, in the title please clarify that the second set of numbers in [#] are standard errors (you need to make this clear…typically this should be parameter estimate and standard error but since this is not labeled there is no way for anyone to know this without guessing)? Also, please change the brackets to parentheses so consistent with Table 5 and standard formatting for this. Please un-bold the [0.626] and please make sure that all alignment of words and numbers in each row are correct…for example, the word “Value-Added” is not in alignment with the numbers in the other columns within that row. This happens elsewhere and needs correction. Also, please create a version of this table that is more understandable to someone not familiar with econometrics. Put the long version as Table A4 at the end and refer readers there for specifics. For the shorter version of Table 4 you should be able to just list the parameter estimate (and significance) which would cut the size of the table in half. Also, what is “_cons”? Also, reduce the length and font size of the footnote and make sure the smaller font size used for footnotes are consistent across ALL tables.

L441 – Get rid of “de”

L443 – “Table” needs to be capitalized

L446-447 – For Table 5, please address the same issues as in prior tables and please create a version of this table that is more understandable to someone not familiar with econometrics. Put the long version as Table A5 at the end and refer readers there for specifics. For the shorter version of Table 4 you should be able to just list the parameter estimate which would cut the size of the table in half.

L496 – Get rid of period after “shops”

L503 – Change to “[35,37-41]” and make sure all repeated citation numbers are in numerical order relative to other repeated citation numbers.

L563 – Change to “Our research suggests that…” since the implications of the research are present tense…if past tense does that imply that the implications are no longer valid?

L585 – Change to “…database allowed us to model…” since the statistical analysis by the time someone is reading the published article is past tense and already completed

L594 – Change to “…located in a city…”

L599-604 – Change to “More generally, we discuss how our findings may consolidate understanding of coffee quality across specialty stakeholders. Our research results suggest the specialty coffee niche in Mexico has some elements required to support values-based supply chains (VBSCs). In this sense, our findings…” so you are deleting the first sentence on VBSC that is redundant to what is being stated.

L608 – Change to “research results could be extended…”

L624-629 – Please change the Appendix to a Supplementary Materials Word file (separate file and you will see MDPI has a template for these as well. This allows the reader to refer to this if they need to by downloading this separate pdf file from the published article pdf. It will also reduce the length of the article. You will have to change the table labels to Table S1, Table S2,

Table S3, Table S4, and Table S5. The other advantage of this is you can make this separate Word file “Landscape” page layout which will be a better format for Table S3 (correlation matrix) so the numbers will now fit and not bump negative numbers to a second row.
